# Construction of Water Corridors for Mitigation of Urban Heat Island Effect

**Jiqing Lin [1], Wufa Yang [2], Kunyong Yu [3,4], Jianwei Geng [3,4] and Jian Liu [3,4,\***

[1] Fujian Chuanzheng Communications College, Fuzhou 350007, China
[2] Fuqing Municipal Bureau of Natural Resources and Planning, Fuzhou 350300, China
[3] College of Forestry, Fujian Agriculture and Forestry University, Fuzhou 350002, China
[4] University Key Lab for Geomatics Technology and Optimized Resources Utilization in Fujian Province, Fuzhou 350002, China
[*] Correspondence: fjliujian@fafu.edu.cn; Tel.: +86-0591-863-92-212

**Abstract:** The urban heat island (UHI) effect is becoming increasingly prominent owing to accelerated urbanization in Fuzhou, affecting the lives of people. Water is an important landscape element that can effectively improve the urban thermal environment. The construction of water corridors has been proven to mitigate the intensity of the UHI effect in Fuzhou. Therefore, we obtained the distribution of a water system in Fuzhou from image data and analyzed temperature watersheds using the inversion of surface temperature to investigate the inner mechanism of the water system influencing the UHI effect. The water system was superimposed with hot spots to obtain cooling ecological nodes and construct water corridors to mitigate the UHI effect. The temperature watershed areas in Fuzhou are: Minhou County (353.77 km$^2$), Changle (233.06 km$^2$), Mawei (137.82 km$^2$), Cangshan (71.25 km$^2$), Jin'an (55.99 km$^2$), Gulou (16.93 km$^2$), and Taijiang (15.51 km$^2$) Districts. Hot spots were primarily located in Changle, Cangshan, Jin'an, Gulou, and Taijiang Districts. The superposition of the water system and temperature watershed yielded 152 cooling ecological nodes, which were concentrated in the Minjiang and Wulong River watershed, with no cooling ecological nodes distributed within the central city. Twenty-five cooling ecological nodes were selected in the hot spot areas, which were primarily distributed in reservoirs, inland rivers, and park water systems. We constructed 12 water corridors, including four, three, two, one, one, and one in the Minhou County, Changle, Mawei, Jin'an, Cangshan, and the Gulou and Taijiang Districts.

**Keywords:** water corridor; urban heat island mitigation; watershed temperature; ecological node





## 1. Introduction

Urban heat islands (UHIs) are a common global phenomenon [1–3] that cause urban climate change [4], increase the frequency of extreme weather [5], affect the material cycle [6] and energy metabolism [7,8], intensify the spatial distribution of thermal environment [9], increase the morbidity of residents, threaten human health [10], and pose serious challenges to the lives of people. During the period 1951–2021, the average surface temperature increased at a rate of 0.26 °C per decade in China, which is much higher than the global temperature increase rate. Therefore, UHI mitigation has become crucial [11], and researchers agree that ecological approaches should be used to address the negative outcomes of the UHI effect [12–14].

Water is an important natural landscape for UHI mitigation [15,16]. Rivers can reduce the surface temperature by 3–5 °C [17], and the cooling distance is similar to the river width [18]. Water systems are better in summer compared with autumn and winter [19], and surface water systems have a better cooling effect than linear water systems [20]. Therefore, the construction of water corridors is important for alleviating the UHI effect and ensuring the safety and stability of urban ecosystems.

Water systems can be extracted through field research by using land use types and remote sensing image data. Water systems comprise the interconnected rivers of various types, sizes, and levels [21]. Field research is time-consuming and labor-intensive, and it is difficult to visually analyze the logical relationships between water systems. In recent years, an increasing number of scholars have extracted water systems by using image data, which can improve efficiency, save in manual labor and material resources, and more accurately express the ecological characteristics of rivers. At present, water corridors are primarily extracted using source identification, resistance surface construction, and corridor extraction methods [22]; however, they deviate from the existing water system paths.

Fuzhou is a coastal city in eastern China, which had many lakes, marshes, and rivers in the early days and now has six major water systems covering 107 inland rivers, including the Baima and Jin'an Rivers, and is an important birthplace of the 21st century Maritime Silk Road [23,24]. In the process of large-scale urbanization, urban surface cover and landscape patterns have changed considerably, the natural form of rivers has been destroyed, ecological units have been fragmented, and important ecological nodes, such as water systems, lack organic connections, as the ecological function of the riverfront zone has been degraded and the discrete water system can hardly exert a cooling effect. In 2007, Fuzhou became one of the four new hot spots in China [25]. Since 2019, the temperature has continued to increase in summer and optimizing the existing water system by linking it in series, leading the lines with the point and surface, respectively, to form a local microclimate, maximizing the cooling effect of the water system, and achieving the alleviation of the UHI in Fuzhou is required. In this study, we extracted the water system and inverse surface temperature directly from image data, identified hot spot areas, simulated the temperature change patterns, obtained the cooling ecological nodes with the help of hydrological, superposition, and hot spot analysis, and subsequently constructed water corridors with a cooling effect. This study attempts to preserve existing water system paths and add new water system paths based on urban hot spots to constitute tandem water corridors.

## 2. Study Area and Methodology

### 2.1. Overview of Study Area

The study area is the central urban area surrounded by Wuhu, Qishan, Lianhua, and Gushan Mountain (Figure 1), which contains Gulou, Jin'an, Taijiang, Mawei, Cangshan, and Changle District, and part of Minhou County (Ganzhe Street, Shangjie, Nanyu, and Nantong Town, etc.), with a total area of 1759.4 $km^2$.

### 2.2. Research Methodology

2.2.1. Remote Sensing Image Inversion of Surface Temperature

The limited distribution of weather stations leads to limitations in meteorological data, which affects the spatial identification of the risk of a heat wave risk [26]. The inversion of land surface temperature (LST) by remote sensing images can quickly and comprehensively obtain the surface temperature. The UHI effect is most obvious in summer (July–September). On 22 September 2019, a remote sensing map of the study area (downloaded from the website of the US Geological Survey) with a cloud cover of <2% was selected and the surface temperature was inversed by the radiative transfer equation method (atmospheric correction method) with high computational accuracy. The formula is as follows:

$$L_\lambda = [\varepsilon B(T_s) + (1 - \varepsilon) L_d]\tau + L_\mu \qquad (1)$$

$L_\lambda$ is the radiation value in the 10th band, $L_d$ is the atmospheric upward radiance, $\varepsilon$ is the surface emissivity, $B(T_S)$ is the thermal radiance of a blackbody at $T_S$ temperature, and $\tau$ is the atmospheric permeability (the related information was obtained from

https://atmcorr.gsfc.nasa.gov, accessed on 22 September 2019). The radiation brightness of a blackbody at temperature T in the thermal infrared band $B(T_S)$. The formula is as follows:

$$B(T_s) = \frac{L_\lambda - L_\mu - \tau(1 - \varepsilon)L_d}{\tau\varepsilon}$$ (2)

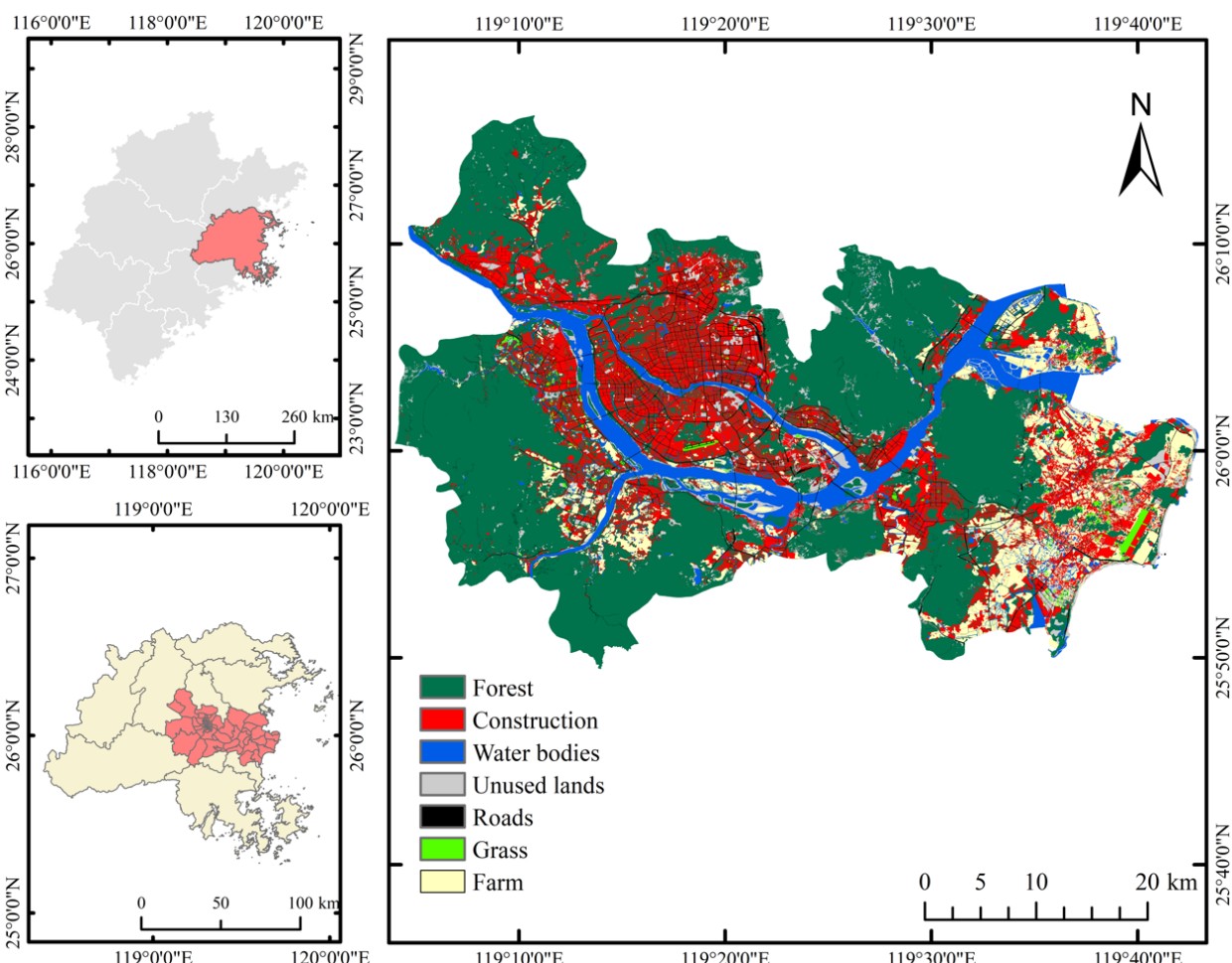

**Figure 1.** Study Area.

The true surface temperature $T_S$ was found according to the inverse function of Planck's formula, whilst $K_1$ and $K_2$ are the values of the 10th band, $K_1$ = 774.89 W/(m² · μm · sr) and $K_2$ = 1321.08 K. As such, the formula is as follows:

$$T_s = \frac{K_2}{\ln(K_1/B(T_s) + 1)}$$ (3)

### 2.2.2. Identification of Cold and Hot Spots of Surface Temperature

The spatial aggregation degree is used to identify the cold spot and hot spot areas of surface temperature [27]. The spatial aggregation of high and low values of surface temperature was measured by the statistical tool Getis-Ord General G. The formula is as follows:

$$Z(G_i^*) = \frac{G_i^* - E(G_i^*)}{\sqrt{Var(G_i^*)}}$$ (4)

$E(G_i^*)$ is the mathematical expectation of $G_i^*$ and $Var(G_i^*)$ is the variance of $G_i^*$ When $Z(G_i^*) > 0$ and significant, it indicates that all values around position $i$ are greater than the mean value and belong to the high-value spatial agglomeration area (hot spot); on the

contrary, when $Z\left(G_i^*\right) < 0$, it indicates that all values around position $i$ are less than the mean value and belong to the low-value spatial agglomeration area (cold spot).

### 2.2.3. Extraction and Classification of Temperature Watershed

We examined the movement trend and path of the surface temperature and used a hydrological analysis model to simulate its movement path and convergence characteristics to obtain the temperature convergence network. The pixel ID code of the surface temperature grid map, whose ID code number has a power of 2, outputs a collection network, and extracts the pixel with higher concentration value to form a temperature convergence network. The temperature convergence classes and area were extracted using the relative magnitude of the convergence flow [28] and the natural breakpoint method was used to classify the temperature watershed into classes 1–4.

### 2.2.4. Extraction and Classification of Water System

Rivers in Fuzhou can be divided into tidal, plain, and mountain rivers, according to their geomorphological and hydrodynamic characteristics [29], among which tidal (Min and Wulong River) and plain rivers (Jin'an and Baima River, etc.) are perennial. Rivers in mountainous areas are mountain streams, which are influenced by terrain fall and cover many important catchment nodes, including depressions and washes. The water volume of these rivers is usually small and rises sharply during the rainy season, forming mountain rapids with intermittent water storage for a short period of time. The water systems in the study area can be divided into two categories: natural and semi-natural [30]. Natural water systems have existed stably here for a long time and comprise rivers, diversions, ponds, lakes, and reservoirs. Tidal and plain rivers are natural water systems that were extracted using 2.5 m precision Google map in 2019. Mountain rivers are semi-natural water systems in which stormwater runoff and catchment areas were identified and extracted through hydrological analysis models. Rainwater runoff and catchment areas were obtained from 2019 digital elevation model (DEM) data (http://www.gscloud.cn accessed on 20 May 2019) with 30 m resolution in Fuzhou. The method of water system classification is the same as the method of temperature watershed classification, and the pixel assignment method is used. The higher the number of tributaries, the higher the classes of water system. The classification of catchments is divided according to the confluence network of the water system. The bigger the catchment area is, the bigger the corresponding catchments class.

### 2.2.5. Construction of Water Corridor

The construction of water corridors is closely related to the selection of ecological nodes. Although there is no unified definition and boundary of ecological nodes, there are important components of ecological networks [31] that are located in the weak areas of ecological corridors [32]. Ecological nodes can be divided into three categories: resource-based strategic [33], structural weak, and structural strategic points [34]. Structural weak points are located at the weak points of ecological corridors, such as the intersection of corridors [35,36], turning points [37], and the intersection of corridors and ecological source sites [38]. We inverted the surface temperature from the remote sensing map, analyzed the cooling ecological nodes inside the water system and the cooling ecological nodes generated by the superposition of the water system and temperature watershed, identified the areas that need cooling through the water system and urban hot spots, added the cooling ecological nodes in the hot spot areas, and connected the ecological nodes with the water system in series to construct the water corridors.

## 3. Results

### 3.1. Analysis of Surface Temperature

The highest surface temperature in the study area was 53.91 °C and the lowest was 20.05 °C (Figure 2). The thermal environment in Fuzhou exhibited a high temperature in the center of the city and a low temperature in the periphery. The central urban area

had a high surface temperature because of its high construction density, little vegetation cover, and the concentration of a large number of industrial and residential areas, medium and large commercial centers, CBD business districts, airports, municipal roads, and other infrastructure. Furthermore, Changle and Yixu Airports had high temperatures, which are heat sources in addition to the large hard pavement area absorbing solar radiation. The surface temperature was low in areas, such as woodlands, parks, rivers, lakes, wetlands, and farmlands.

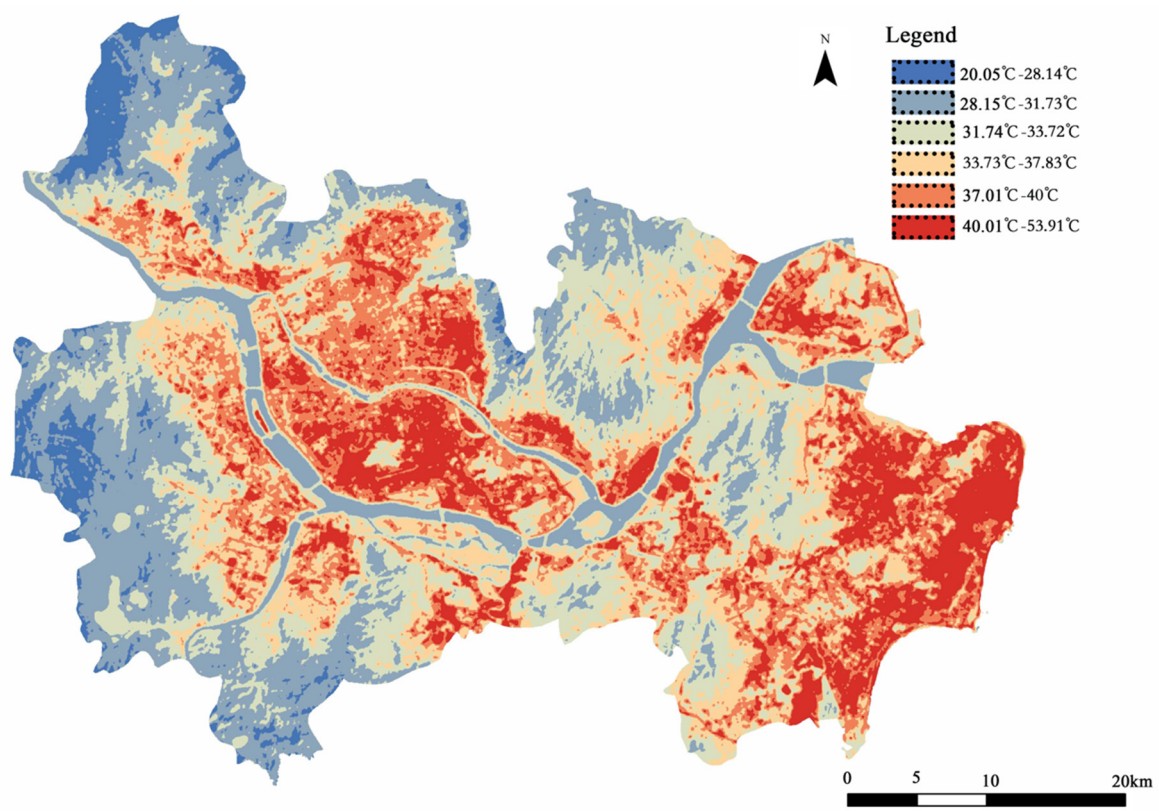

**Figure 2.** Surface temperature.

*3.2. Analysis of Cold and Hot Spots*

We identified various cold and hot spots in Fuzhou (Figure 3). The hot spots in Cangshan District were distributed in areas other than Jianxin Town and the Sanjiangkou Zone. The hot spots in Gulou District were distributed in areas other than parks and green areas, such as Jinniushan, Xihu–Zuohai, and Pingshan Parks. The hot spots were distributed throughout the entire area of Taijiang District, which has less vegetation and water bodies and more construction land. The hot spots in Jin'an District were distributed in the northern and primary urban areas. Many hot spots were distributed in the residential areas of East Riverside, Tingjiang Town, and Lanqi Island in Mawei District. The hot spots were large in the primary urban and airport areas and other township built-up areas in Changle District. The hot spots in Minhou County were distributed in the High-tech Zone, Ganhong Road residential area, part of Ganzhe Town, Nanyu residential area, Nantong Township, and Qingkou Industrial Zone.

*3.3. Analysis of Temperature Watershed*

The temperature watershed was divided into four levels based on the natural break-points (Figure 4). The larger the value of the level was, the higher the degree of the regional temperature pooling was, and the more obvious the trend of movement from high to low temperature areas was. The first-level temperature watershed was distributed in the Minjiang River. The second-level temperature watershed was primarily distributed in the

Minjiang River and its path branches, partly concentrated in the Minjiang and Wulong Rivers and partly converged in Minjiang–Changle Marina New Town. The third-level temperature watershed converged in natural woodlands, such as Wuhu, Qishan, Lianhua, and Gushan Mountains. The fourth-level temperature watershed was the most densely distributed and concentrated in the built-up area, and the overall trend was consistent with the characteristics of surface temperature movement.

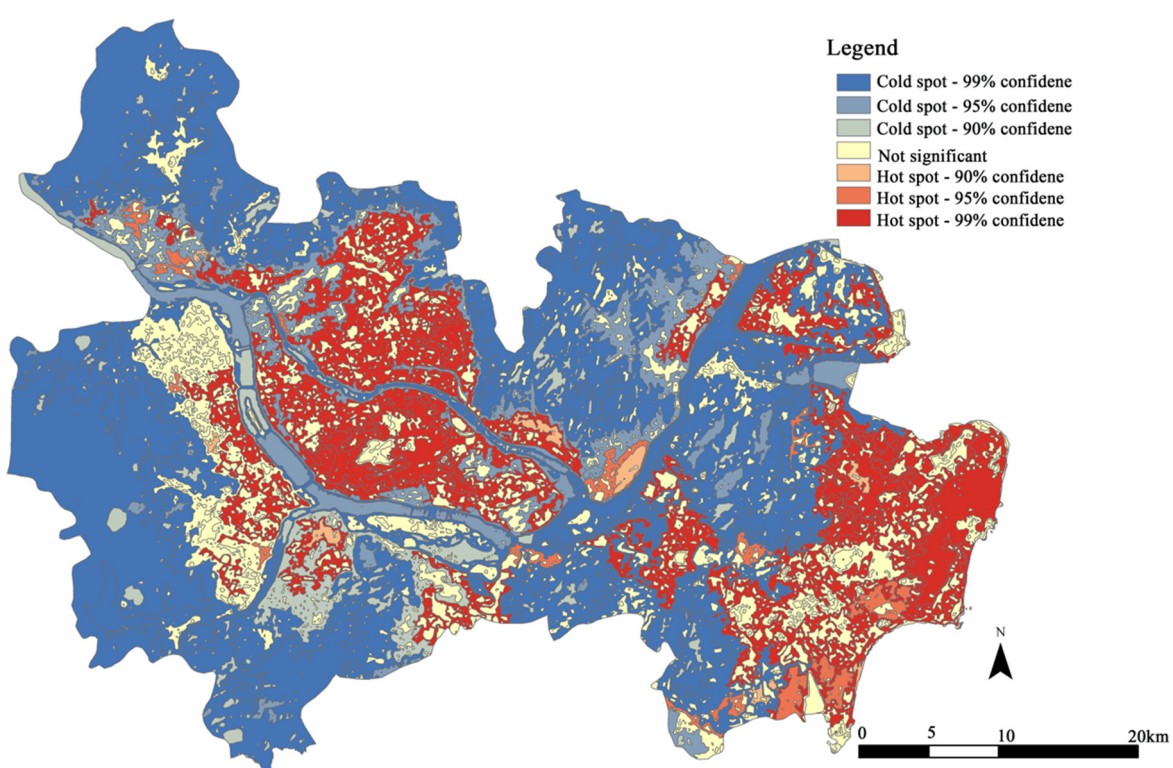

**Figure 3.** The cold spots and hot spots spatial distribution.

The spatial distribution of temperature watershed (Table 1) exhibited that the total sizes of the temperature watersheds were in the following order: Minhou County (353.77 km$^2$) and Changle (233.06 km$^2$), Mawei (137.82 km$^2$), Cangshan (71.25 km$^2$), Jin'an (55.99 km$^2$), Gulou (16.93 km$^2$), and Taijiang Districts (15.51 km$^2$). Among them, the first-level temperature watershed areas were in the following order: Minhou (322.19 km$^2$), Changle (201.10 km$^2$), Mawei (92.90 km$^2$), Cangshan (50.71 km$^2$), Jin'an (48.47 km$^2$), Gulou (16.93 km$^2$), and Taijiang (10.01 km$^2$). The size of the second-level temperature watershed area was in the order: Minhou (31.58 km$^2$), Cangshan (20.54 km$^2$), Mawei (18.45 km$^2$), Changle (11.70 km$^2$), Jin'an (7.52 km$^2$), Taijiang (5.50 km$^2$), and Gulou (0 km$^2$). The third and fourth level temperature watersheds were only distributed in Mawei (15.70 km$^2$ and 10.77 km$^2$, respectively) and Changle (18.70 km$^2$ and 1.56 km$^2$, respectively).

The spatial distribution of the temperature watershed demonstrated that the trend of the temperature movement flowed from high temperature areas in the central city to low-temperature areas, such as water bodies, urban parks, and agricultural and forestry land in the suburbs, forming a difference between the high and low pressure, which is conducive to the exchange and circulation of heat energy inside and outside the city.

### 3.4. Analysis of Water System

The natural water system (Figure 5) had clear spatial distribution characteristics, with a dense distribution of water systems and a well-developed network of inland rivers. The primary river networks of the Minjiang and Wulong Rivers were dominated by linear water systems, whereas the rest of the water systems mostly comprised primary branch water

systems and inland rivers. The distribution of the surface water systems was smaller and more scattered in terms of area.

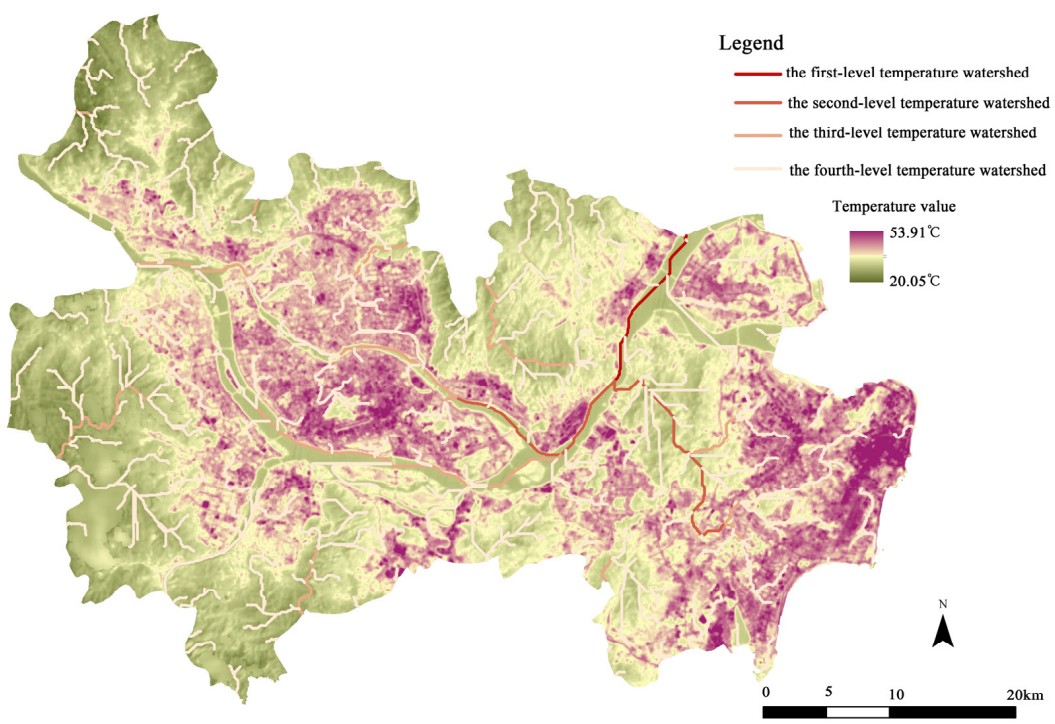

**Figure 4.** Spatial distribution of temperature watershed.

**Table 1.** Spatial distribution of temperature watershed.

| Temperature Watershed Level | Administrative Regions (Unit: km$^2$) | | | | | | |
|---|---|---|---|---|---|---|---|
| | Cangshan | Gulou | Jin'an | Mawei | Minhou | Taijiang | Changle |
| First-level | 50.71 | 16.93 | 48.47 | 92.90 | 322.19 | 10.01 | 201.10 |
| Second-level | 20.54 | 0 | 7.52 | 18.45 | 31.58 | 5.50 | 11.70 |
| Third-level | 0 | 0 | 0 | 15.70 | 0 | 0 | 18.70 |
| Fourth-level | 0 | 0 | 0 | 10.77 | 0 | 0 | 1.56 |
| Total | 71.25 | 16.93 | 55.99 | 137.82 | 353.77 | 15.51 | 233.06 |

Table 2 lists the natural water system in Fuzhou, exhibiting that the total area of the linear water system was 129.92 km$^2$, among which the largest area (51.19 km$^2$) was in Minhou County, primarily including the Wulong, Minjiang, and Dazhang Rivers and their tributaries—followed by that in Mawei District (27.05 km$^2$) and those of Changle and Cangshan Districts with areas of 24.09 km$^2$ and 21.86 km$^2$, respectively, which is less than in the primary urban area. The primary urban areas in Taijiang, Jin'an, and Gulou Districts were only 3.41 km$^2$, 1.33 km$^2$, and 0.99 km$^2$, respectively. The total area of the surface water system was 28.06 km$^2$. The largest area was in Changle District (12.85 km$^2$), followed by those in Mawei (6.96 km$^2$) and Minhou Districts (6.33 km$^2$), and several other districts were smaller. The primary urban area had a high degree of land intensification and fewer reservoirs, lakes, and farmlands. The total area of the water system in the administrative districts were: Minhou County (57.52 km$^2$) > Changle District (36.94 km$^2$) > Mawei District (34.01 km$^2$) > Cangshan District (22.75 km$^2$) > Taijiang District (3.42 km$^2$) > Jin'an District (1.96 km$^2$) > Gulou District (1.38 km$^2$).

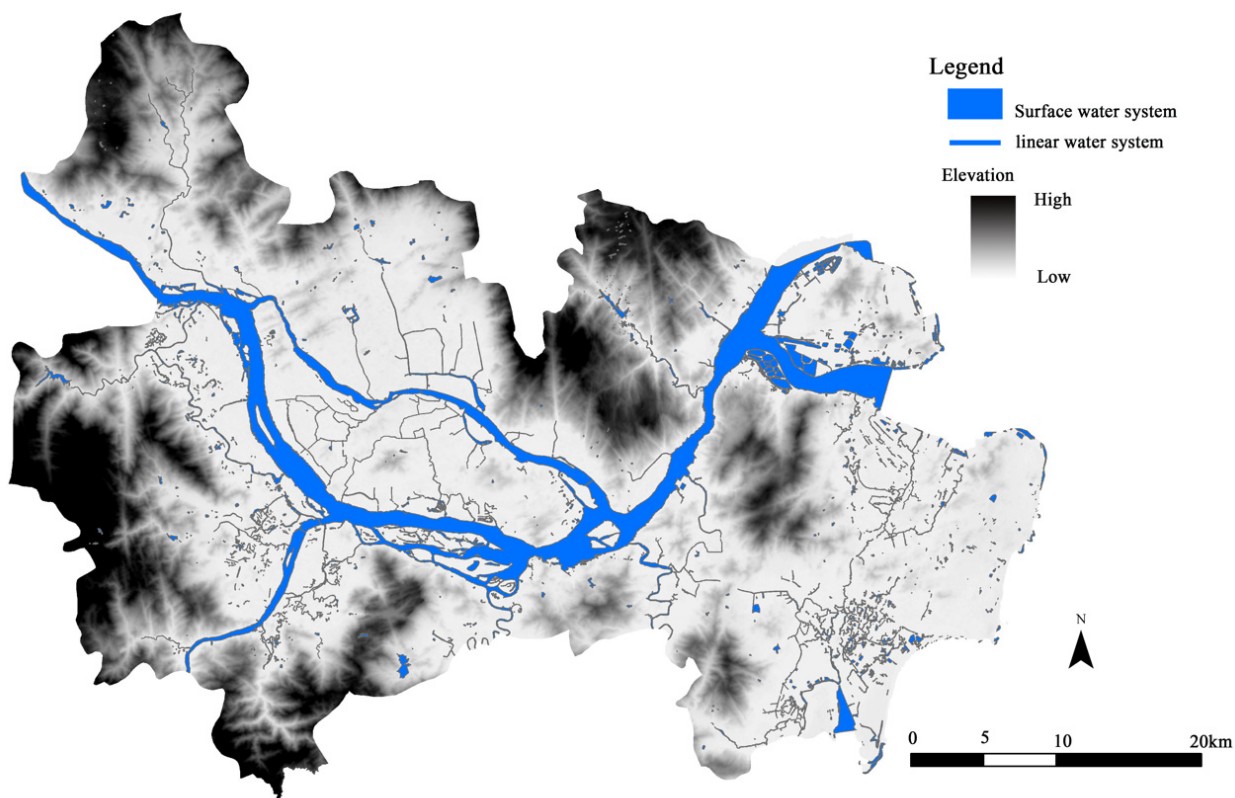

**Figure 5.** Natural water system in Fuzhou, China.

**Table 2.** Natural water system in Fuzhou, China.

| Water System Type | Administrative Regions (Unit: km$^2$) | | | | | | | |
|---|---|---|---|---|---|---|---|---|
| | Cangshan | Gulou | Jin'an | Mawei | Minhou | Taijiang | Changle | total |
| Linear water system area | 21.86 | 0.99 | 1.33 | 27.05 | 51.19 | 3.41 | 24.09 | 129.92 |
| Surface water system area | 0.89 | 0.39 | 0.63 | 6.96 | 6.33 | 0.01 | 12.85 | 28.06 |
| Total | 22.75 | 1.38 | 1.96 | 34.01 | 57.52 | 3.42 | 36.94 | 157.98 |

The semi-natural catchments in Fuzhou (Figure 6) were divided into five classes, according to the natural breakpoint method. The larger that the value of the class was, the greater the capacity of the rainwater pooling was, and the more favorable the formation of catchments was. As shown in Figure 6, the darker the color was, the easier it was to form catchments. The largest catchment areas were in the Minjiang and Wulong River watersheds because rainwater from the surrounding mountainous areas converges into the inland rivers and tributaries, which eventually converge in the Minjiang and Wulong Rivers.

According to the administrative division in Fuzhou, the total areas of rainwater catchment (Table 3) were in the following order: Minhou County (660.96 km$^2$) > Changle (406.86 km$^2$) > Mawei (234.83 km$^2$) > Cangshan (147.29 km$^2$) > Jin'an (107.94 km$^2$) > Gulou (37.28 km$^2$) > Taijiang District (18.48 km$^2$). Minhou County and Changle and Mawei Districts had high values, primarily because these areas are larger than other regions and have more catchment areas formed by the confluence of mountains within their jurisdictions. In addition, the catchment areas of these districts occupy most of the Minjiang and Wulong Rivers. The catchment areas of Gulou and Taijiang Districts were much smaller than those of other jurisdictions, with a high degree of land intensification, the largest construction sites, and a smaller catchment formed by park water bodies and inner water systems.

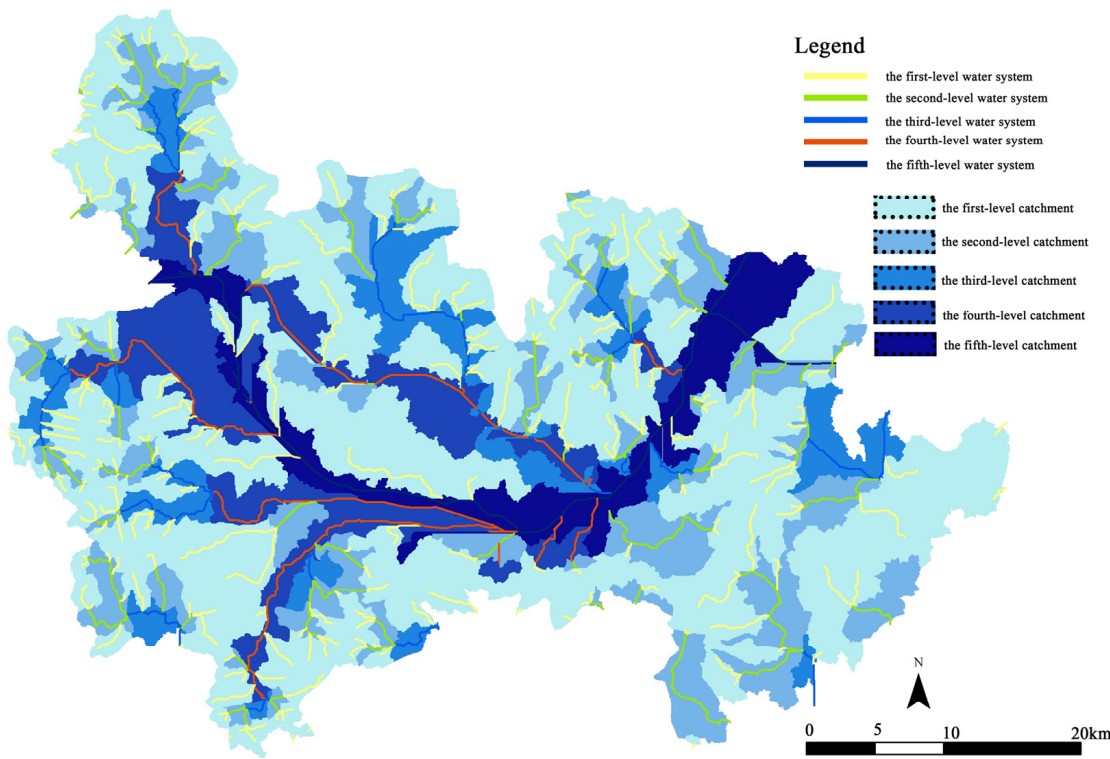

**Figure 6.** Semi-natural catchment spatial distribution.

**Table 3.** Catchment area of administrative regions.

| Administrative Regions | Area (Unit:km²) | | | | | |
|---|---|---|---|---|---|---|
| | First-Level Catchment | Second-Level Catchment | Third-Level Catchment | Fourth-Level Catchment | Fifth-Level Catchment | Total |
| Jin'an | 64.74 | 13.42 | 28.05 | 1.73 | 0 | 107.94 |
| Taijiang | 9.14 | 0.22 | 1.15 | 7.97 | 0 | 18.48 |
| Cangshan | 70.26 | 8.25 | 9.68 | 27.81 | 31.29 | 147.29 |
| Gulou | 20.68 | 3.01 | 5.28 | 8.31 | 0 | 37.28 |
| Mawei | 125.66 | 41.5 | 15.65 | 13.49 | 38.53 | 234.83 |
| Minhou | 332.79 | 124.06 | 54.4 | 120.9 | 28.81 | 660.96 |
| Changle | 218.61 | 123.7 | 30.66 | 6.53 | 27.36 | 406.86 |

The natural and semi-natural water systems were spatially superimposed to form a water system (Figure 7). From the spatial distribution of the water system, the primary watershed of the Minjiang and Wulong River tandem tributary water system formed a buffer zone from the water system to spread outward. The western and central catchment areas were relatively dense and covered a wide area, whereas the eastern catchment areas were relatively sparse and covered less area, with an overall east–west direction.

*3.5. Analysis of Cooling Ecological Node*

3.5.1. Cooling Ecological Nodes in the Water System

Sixty-nine cooling ecological nodes were generated in the water system (Figure 8), among which 1, 1, 2, 10, 14, 4, and 37 were present in Gulou, Taijiang, Jin'an, Changle, Cangshan, and Mawei Districts, and Minhou County, respectively, primarily located in the Dazhang River, inland rivers, and large reservoirs. The cooling ecological nodes in the central city were concentrated in the Minjiang and Wulong River watersheds, and no ecological nodes appeared within the city, implying that the ecological function of the water system in the central city was relatively singular.

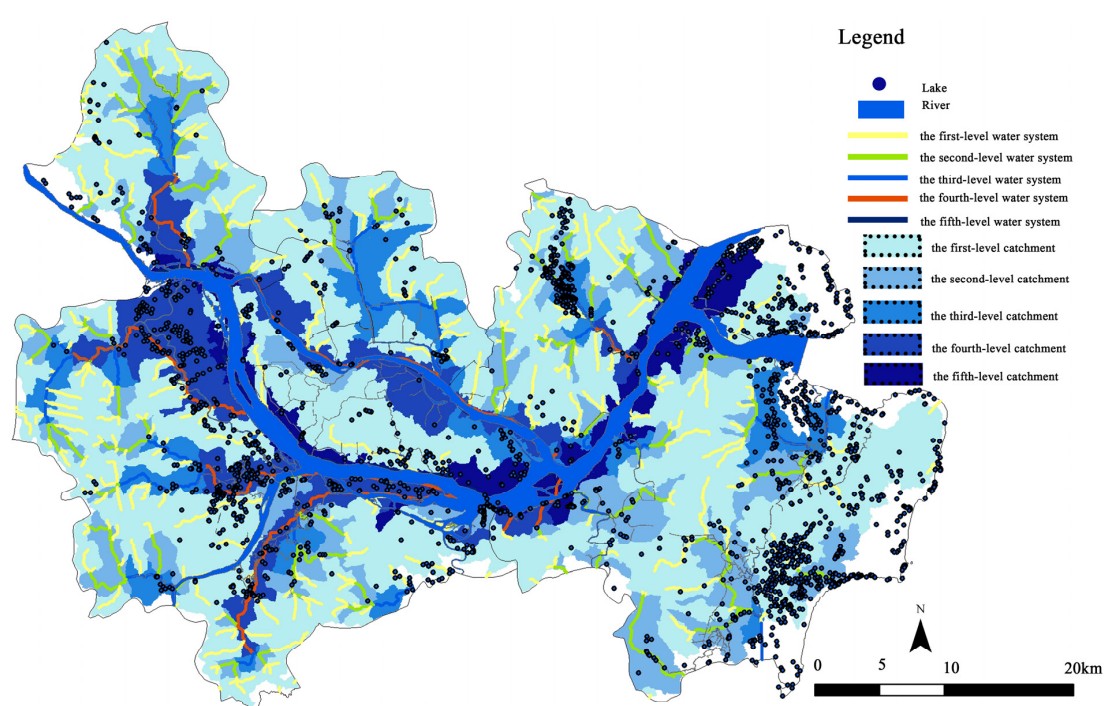

**Figure 7.** Water systems in Fuzhou, China.

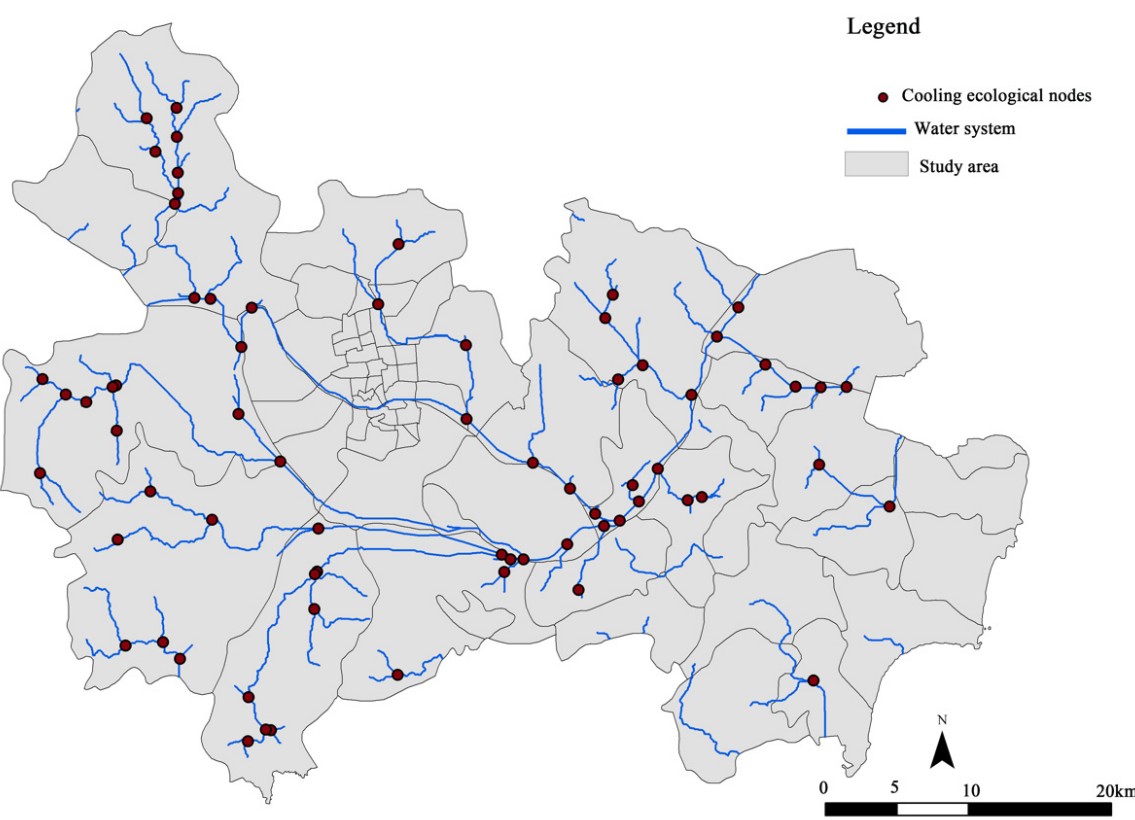

**Figure 8.** Distribution of the cooling ecological nodes of water system.

3.5.2. Cooling Ecological Nodes in Superimposed Water System and Temperature Watershed

Through the spatial analysis function of GIS software, the superposition of water systems and temperature watersheds yielded 152 nodes (Figure 9): 47, 31, 28, 25, 14, 4, and 3

in Minhou County and Mawei, Cangshan, Changle, Taijiang, Jin'an, and Gulou Districts, respectively. These nodes are cooling ecological nodes, and they were concentrated in the Minjiang and Wulong River watersheds and their sub-watersheds. There were no cooling ecological nodes in the central city. The cooling effect of water systems on the UHIs was not obvious in the central city, and more cooling nodes are needed to improve the cooling benefits of water corridors.

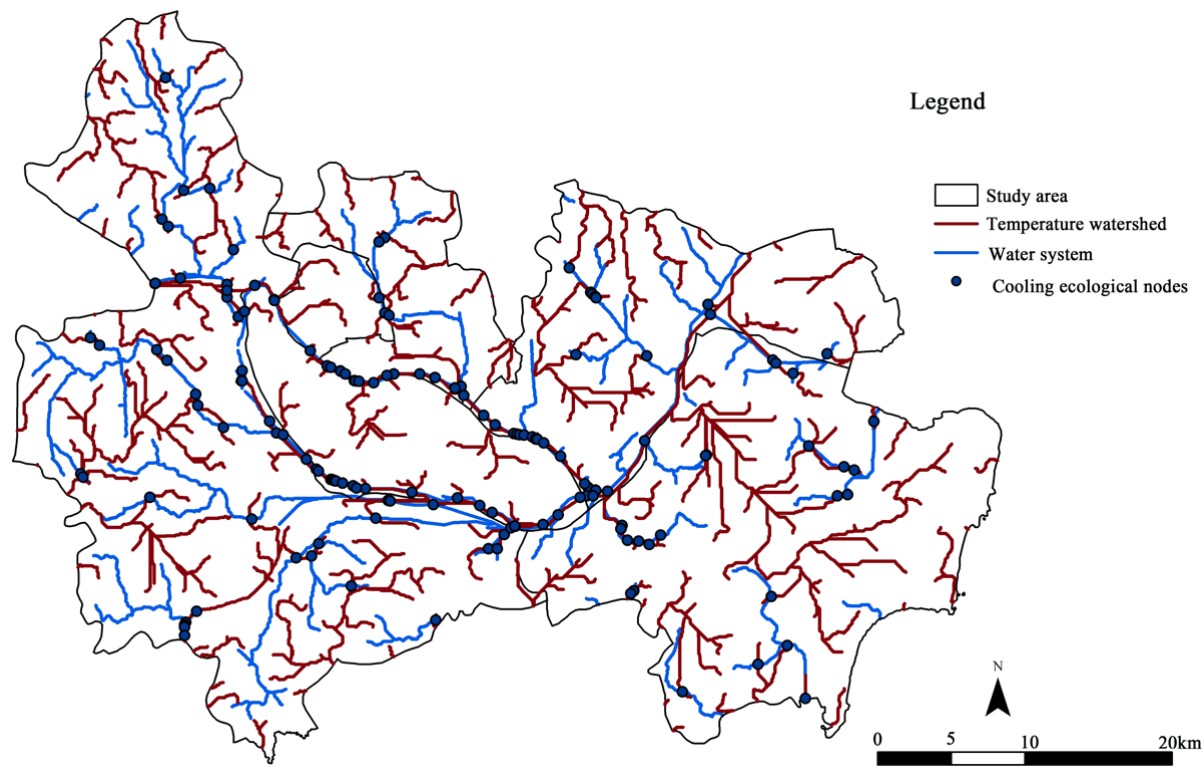

**Figure 9.** Water system and temperature watershed superimposed map.

### 3.5.3. Cooling Ecological Nodes in Superimposed Water Systems and Hot Spots

Twenty-two high temperature loopholes were identified in superimposed water systems and hot spots (Figure 10), which were concentrated in the central city, town areas, industrial zones, and airports. Seven were in Minhou County, including Ganzhe, Jingxi, Nantong, and Nanyu Towns, Shangjie University Area, and High-tech and Qingkou Industrial Zones. Six were in Cangshan District, including the Jinshan Industrial Zone, Wanda Square, Olympic Sports Center, Sanchajie, Baihuting, and Huangshan. Three were in Jin'an District, including the North Railway Station, North and East Taihe Plaza, and Logistics Zone. Three were in Mawei District, including Kuiqi, Kuai'an, Luoxing Old Town, and Langqi Island residential area. Two were in Gulou District, including Dongjiekou and Wuyi Square. One was in Taijiang District, namely Chating. Twenty-five cooling ecological nodes, including reservoirs, inland rivers, and parks were in the hot spot areas (Table 4).

### 3.6. Analysis of Cooling Water Corridor

According to the planning documents for Fuzhou, such as Fuzhou General Planning (2010–2020), Fuzhou New District General Planning (2015–2020), Fuzhou Urban Green Space System General Plan (2011–2020), and Fuzhou Urban Comprehensive Transportation Planning (2011–2020), 25 cooling ecological nodes form 12 water system paths that converge in Minjiang and Wulong Rivers (Figure 11, Table 5), including four in Minhou County, three in Changle District, two in Mawei District, one in each Cangshan and Jin'an Districts, and one in the Gulou and Taijiang Districts, considering the characteristics of water flow.

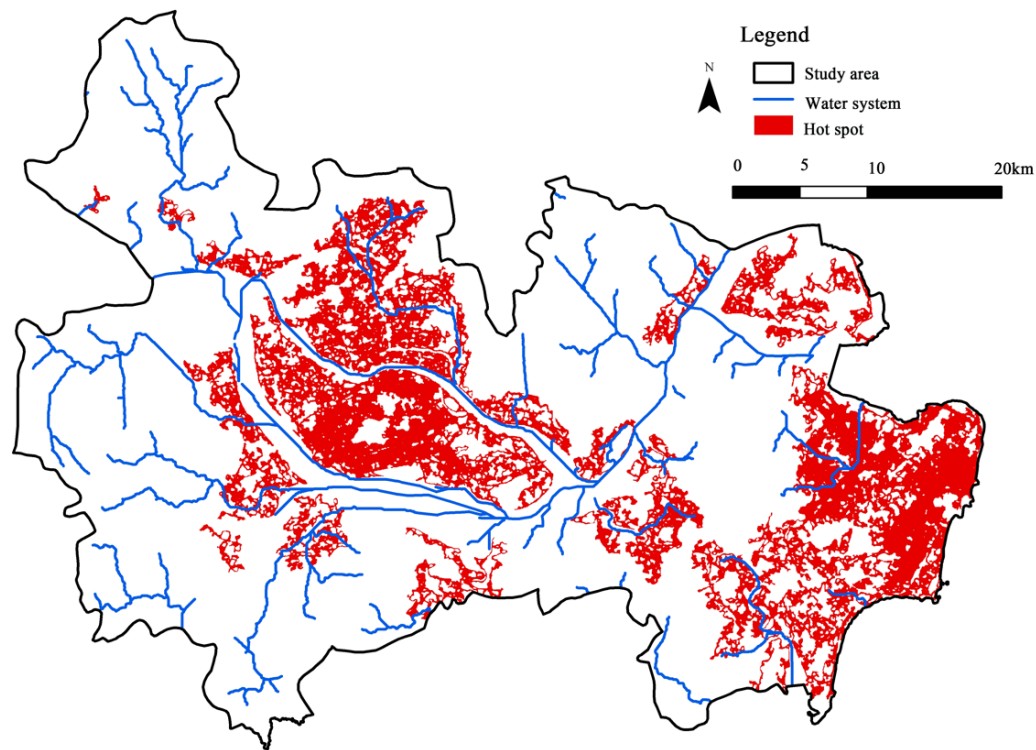

**Figure 10.** Water system and hot spot superimposed map.

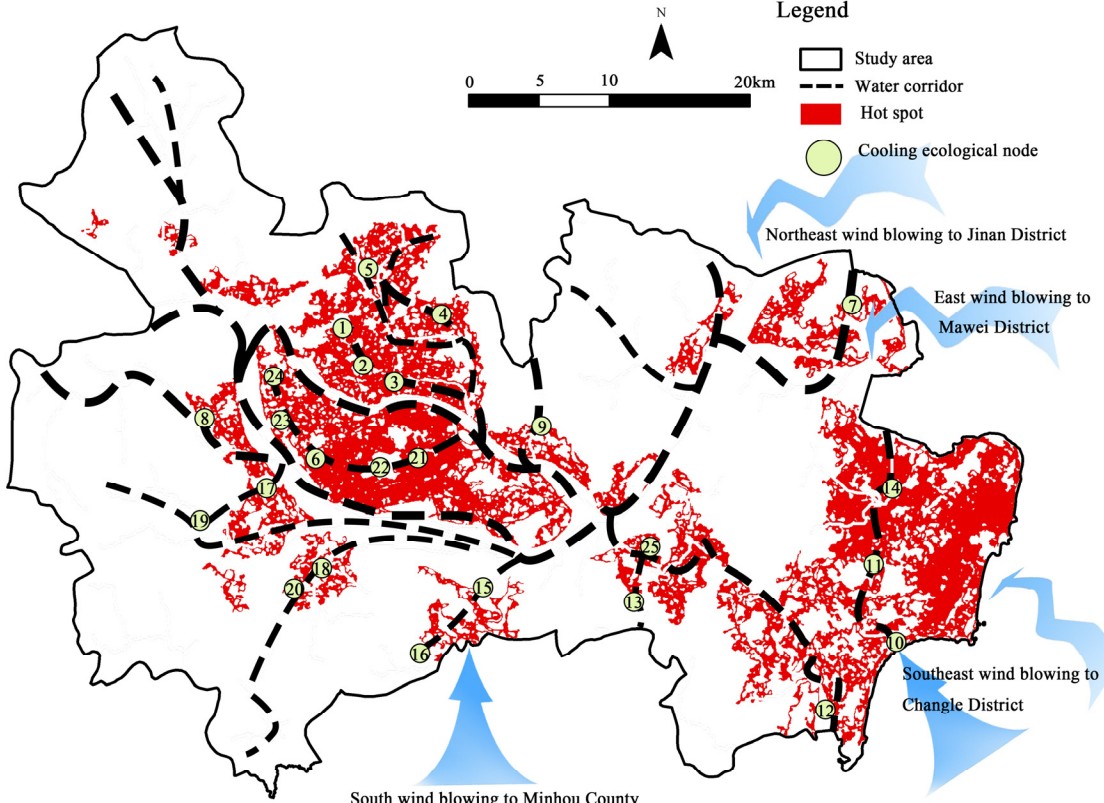

**Figure 11.** Cooling water corridor in Fuzhou, China.

**Table 4.** New cooling ecological nodes.

| Administrative Regions | Number | Cooling Ecological Nodes District Location (No.) |
|---|---|---|
| Gulou | 2 | Xihu Park (1), Liminghu Park (2); |
| Taijiang | 1 | Jin'an River and Guangming Port Intersection Area (3); |
| Jin'an | 2 | Helin Ecological Park (4), Qintinghu Park (5); |
| Cangshan | 5 | Feifeng Mountain Olympic Theme Park (6), Baihuting(21), Gao Gai Mountain Park (22), Jinshan Park (23), Zhenban Village Mountain (24); |
| Mawei | 2 | Langqi Xingfu Reservoir (7), Moxi Scenic Area (9); |
| Changle | 6 | Changle mussel Park (10), Caizhai Village (11), Wenwusha Reservoir (12), Yutian Town (13,) Wenling Village (14), Chang'an Park (25); |
| Minhou | 7 | Tajiaozhou Wetland Park (15), Shigu Mountain (16), Xiyuanjiangkou (17), Zeyang Village (18), Qishan Qipangshi Scenic Area (19), Dazhangxi Nantong Shangzhou Park (20), Qi'an Village (8). |

**Table 5.** Optimized corridor paths.

| Administrative Regions | Path Type | Newly Generated Corridor Paths |
|---|---|---|
| Gulou Taijiang | Tandem | 1—Baima River Park—2—3—Minjiang River and other waterways along the way; |
| Jin'an | Tandem | 5—North Area—Xindian Creek—4—Guangming Port—Aofeng Zone—Minjiang River and other waterways along the way; |
| Cangshan | Tandem | 21—22—6—Jinshan residential Area—23—Jinshan Industrial Zone—24; |
| Mawei | Tandem Optimization | (1) Minjiangkou–Longtai Village—7—Wuzhuang Village along the water system; (2) Minjiang—Moxi—Kuiqi residential area—9; |
| Changle | Tandem Tandem Tandem | (1) 10—11—14—estuary (11—14 there is a break in the flow); (2) 12—Changle Lianhua Mountain—25—Minjiang River (12—25 there is a break in the flow); (3) 13—25—Minjiang River (13—25 there is a break in the flow); |
| Minhou | Tandem Tandem Optimization Optimization | (1) Wulong River—15—Taojiang River—Shenhai Expressway—16(15—16 there is a break in the flow); (2) 19—17—Wulong River; (3) 8—Xiyuan River—Shangjie University Area—Wulong River (optimized on the basis of the original); (4) 20—18—Wengong River—Wulong River (optimized on the basis of the original); |

## 4. Discussion

In the study area, there are obvious spatial differences in the UHIs, with high surface temperatures in the central city and Changle District and hot spot areas exhibiting a patchy distribution. The high concentration of buildings, dense road networks, and small proportion of water bodies and green areas in the central urban area led to higher surface temperatures, which is consistent with the results of previous studies [39,40]. Changle District is located in an estuary, and the sea breeze can effectively alleviate the UHI; however, the natural landscape decreases and the construction of infrastructure and airport increases in the expansion of the city to the east, and the region demonstrates a trend of concentrated and extensive distribution of hot spots. Therefore, future urban construction should coordinate the ratio of the various types of land.

The surface temperature only reflects the spatial structure characteristics of temperature and not the movement trend of temperature from high- to low-temperature areas. The temperature watersheds simulated by the hydrological analysis model through the study are consistent with the cooling pattern of the water system in the study area. Therefore, the addition of cooling ecological nodes along the movement trajectory of the temperature watersheds can not only improve the connectivity of the path network but also promote energy exchange between high- and low-temperature areas, effectively reducing the UHI intensity.

The water system in the study area comprised natural and semi-natural water systems. The semi-natural water system compensated for the poor connectivity of the natural water system up to a certain extent. A large amount of rainwater in the urban area is dissipated through the rainwater catchment area, cutting down the flood peaks formed by heavy rainfall, such as typhoons and heavy rains, thus avoiding the occurrence of urban flooding due to excessive rainwater accumulation. The wider the coverage of the water system, the greater the mitigation of the UHI effect, and the more obvious the ecological benefits. Semi-natural water systems were obtained from the DEM data to collect rainwater, whereas urban hot spots were mostly located in areas with dense populations, high and dense buildings, industrial zones, commercial complexes, airports, railway stations, other gray infrastructures, and large amounts of anthropogenic heat sources. This is not directly related to the DEM data, which is the reason for the lack of cooling ecological nodes in the hot spots.

The identification of cold and hot spots is beneficial for the analysis of the spatial distribution of temperature in Fuzhou and provides a basis for quantitative research on the benefits of UHI mitigation. Although urban rivers are densely distributed, the surface temperature of the main city is high and the UHI is large in summer. The water system can reduce the surface temperature through evaporative cooling [41] and the formation of a local microclimate [15]. However, the water system can only affect its surrounding areas, and its effect on UHI mitigation is limited. The water system did not effectively cover the hot spot area, resulting in a high surface temperature and an obvious UHI effect. By adding cooling ecological nodes to the hot spot area, the spatial distribution of the water corridor can be enhanced, and the connectivity of the water corridor can be improved.

The purpose of the water corridor construction is to mitigate the UHIs by ecological methods and improve the connectivity of water systems by connecting scattered and disconnected water bodies to inland rivers and eventually to the Wulong or Minjiang Rivers through both tandem and optimization. Connectivity is the process of material and energy transfer in water with the water flow [42]. High connectivity and more loops between ecological nodes are conducive to migration and energy exchange among organisms, promoting the role of various functional flows in the study area and significantly contributing to biodiversity conservation and ecological effectiveness. Nine of the twelve new water systems are in tandem (Table 5), in which the paths are formed by the cooling ecological nodes 1–2–3. The water system paths are located in the Taijiang and Gulou Districts, with the smallest and largest administrative areas of 18.17 km$^2$ and of 37.27 km$^2$ in Taijiang and Gulou Districts, respectively. The two districts are connected, thus integrating the water systems into one path and improving their connectivity, which is conducive to enhancing the cooling effect of the water corridor. In the Changle District, all three water systems have a disconnected flow, which affects their cooling effect. The cooling effect of deep water is better than that of shallow water [43]; however, shallow water also has the effect of cooling the surrounding temperature [44]. Combined with the land use type, the disconnected water system is connected by excavating ditches and other means to stop artificially cutting the water system. In addition, the cooling effect of water systems is influenced by the area, width, shape, form, and surrounding environment [45]. For the three optimized water systems (one and two in Mawei and Minhou Districts, respectively), the paths can be enhanced by adjusting the area and morphology of the water bodies to amplify the intrinsic effect of the water system, thus improving the cooling effect of the water corridor.

The cooling effect of water corridors is better than that of plants [46,47] and wind; however, in areas where the concentration of the hot spot distribution is constrained by the type of land use and the introduction of large water systems is not allowed, cooling can be achieved by combining important natural landscape greenery [48,49]. When the green area is larger, the cooling effect is more significant [50]. In Cangshan District, the water corridor path comprising ecological nodes 21–22–6–23–24 can reduce the surface temperature of the area, but the building land in Cangshan District is 58.99 km$^2$, accounting for 40.48% of its total area, with a large UHI intensity. The area of the water bodies is 22.44 km$^2$ in

Cangshan District, accounting for 15.40% of its total area, which can only partially relieve the UHI. If the cooling effect of greenland and grassland (37.66 km$^2$) is considered, green spaces, such as parks and residential areas, linked in series, can improve the cooling effect of the corridor. The prevailing wind direction in Fuzhou is mainly southeasterly in summer, while the prevailing wind direction in winter is northwesterly, with an annual average wind speed reaching 32.2 m/s (Figure A1). Due to the influence of the natural topographic and geomorphological features of the special estuarine basin in Fuzhou, the Minjiang River inlet and the southeast coastal zone act as import ports for the circulation of sea breeze. A corridor constructed with the interaction of the water system and ventilation can greatly alleviate the UHI intensity of the airport, Heshang, Jinfeng, and Wenling Towns, as well as other areas.

This study has important implications for improving the ecological quality of the environment; however, it also has research limitations. Future research should focus on the two following aspects: First, the selection of cooling ecological nodes must be optimized. The current selection of cooling ecological nodes is divided into two categories, one of which comprises the cooling ecological nodes identified by software, such as: the intersection within the water system, the intersection generated by the superposition of the water system and the temperature watershed; the other comprises the cooling ecological nodes identified by human–computer interaction between the hot spot and the water system. Although the current selection of cooling ecological nodes combines hot spots, existing site topography, and relevant design information, these are subjective assumptions. The purpose of this study was to optimize the integration of water corridors using the existing water system resources. In the future, relevant parameters should be used to quantitatively identify the cooling ecological nodes, improve their accuracy, and enhance their utility. Second, setting the width of the water corridor is difficult [51], and scholars use different methods, such as the ant colony algorithm, minimum cost, and other methods, to determine the width of the ecological corridor [52]. The corridor width results obtained by different algorithms are different. The design of the width of the water corridor should be adapted to local conditions, and this study only considered the path of the water corridor in Fuzhou, which does not involve the width of the corridor. In the future, the actual situation should be combined to further improve the accuracy of the corridor width.

## 5. Conclusions

In this study, the surface temperature in Fuzhou was inferred by using the radiative transfer equation method based on remote sensing and GIS tools, and the water system and temperature watershed maps were superimposed using the hydrological analysis model to obtain the optimized map of water corridors for mitigating the UHI effect. The primary findings of this study are as follows.

1. The surface temperature inversion map exhibiting the overall spatial distribution of the thermal environment in Fuzhou had a trend of high and low temperatures in the central city and periphery, respectively. The total area of temperature watersheds was obtained using the flow trend of temperature in six districts and one county: Minhou County (353.77 km$^2$), Changle (233.06 km$^2$), Mawei (137.82 km$^2$), Cangshan (71.25 km$^2$), Jin'an (55.99 km$^2$), Gulou (16.93 km$^2$), and Taijiang Districts (15.51 km$^2$). The temperature watersheds were divided into four levels: the first-level temperature watersheds were distributed in the Minjiang River watershed; the second-level temperature watersheds were partly concentrated in the Minjiang and Wulong River watersheds and partly converged in the direction of the Minjiang River–Changle District Marina New Town; and the third- and fourth-level temperature watersheds were only distributed in Mawei and Changle Districts.

2. The superposition of the water system and watershed temperature yielded 152 cooling ecological nodes, which were primarily distributed in the Minjiang and Wulong Rivers. Twenty-five new cooling ecological nodes were added through the hot spot and water system overlay. Combining the terrain, water system collection, and related design information, 12 new water system paths were added, including four in Minhou County,

three in Changle, two in Mawei, one in Jin'an, one in Cangshan, and one in the Gulou and Taijiang Districts.

3. The constructed water corridors optimized the distribution of ecological nodes and the coverage of water corridors significantly increased, effectively alleviating the UHI. In areas lacking water resources, plants and wind can be combined to improve the cooling effect of the corridors.

**Author Contributions:** Conceptualization, W.Y., K.Y. and J.L. (Jiqing Lin); methodology, W.Y. and J.L. (Jiqing Lin); software, J.G.; validation, W.Y. and J.L. (Jiqing Lin); formal analysis, J.L. (Jiqing Lin) and W.Y.; investigation, W.Y. and J.G.; resources, K.Y.; data curation, W.Y.; writing—original draft preparation, J.L. (Jiqing Lin); writing—review and editing, J.L. (Jiqing Lin), W.Y., K.Y. and J.L. (Jian Liu); visualization, J.L. (Jiqing Lin); supervision, K.Y. and J.L. (Jian Liu); project administration, J.L. (Jian Liu); funding acquisition, K.Y. All authors have read and agreed to the published version of the manuscript.

**Funding:** This research was funded by the National Natural Science Foundation of China (grant number 31770760) and Fujian Provincial Department of Science and Technology (grant number 2020N5003).

**Data Availability Statement:** The data presented in this study are available upon request from the corresponding author.

**Acknowledgments:** We wish to thank the anonymous reviewers and editors for their detailed comments and suggestions.

**Conflicts of Interest:** The authors declare no conflict of interest.

**Appendix A**

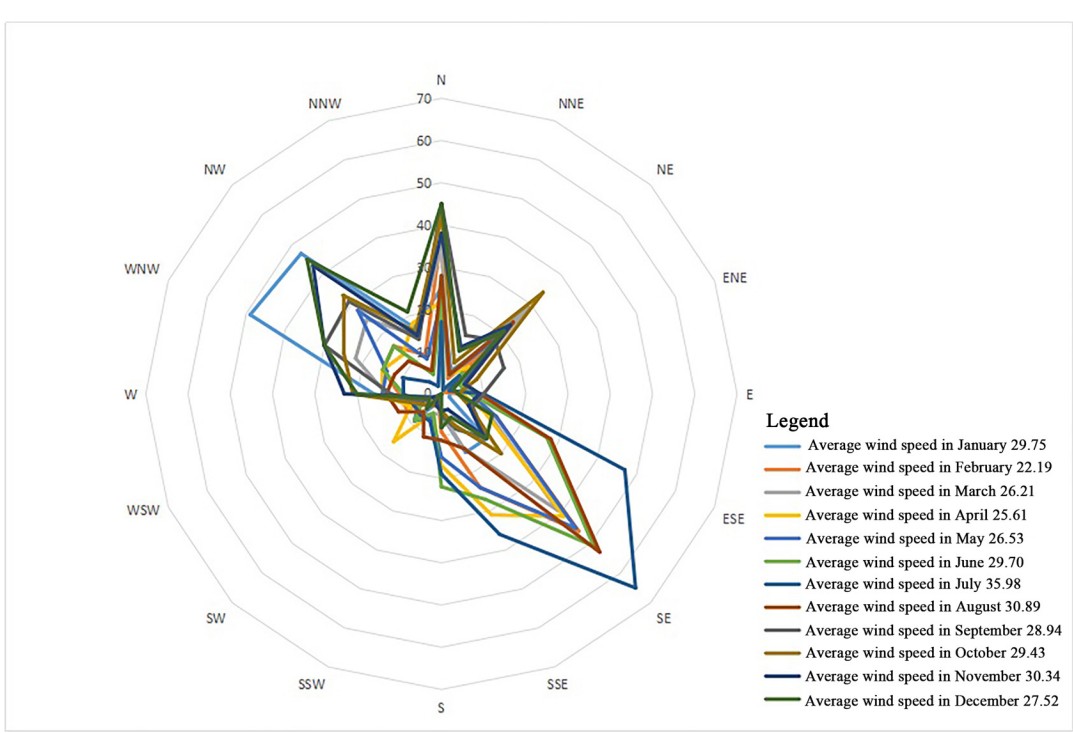

**Figure A1.** Annual wind rose in Fuzhou (m/s) (2015–2019). Data from Fuzhou Climate Station.

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
