# Peer review of "Construction of Water Corridors for Mitigation of Urban Heat Island Effect"

_land, doi:10.3390/land12020308_

Round 1
Reviewer 1 Report
This study assessed the “Construction of Water Corridors for Mitigation of Urban Heat Island Effect”. Urban heating is a global concern, this paper is timely and could offer new insights on water corridors effects on the outdoor environment. The manuscript is generally well written and easy to understand. I suggest that the authors should revise the manuscript incorporating the following comments and suggestions into an updated version.
1. How the authors distinguished various temperature watersheds and computed their areas.
2. On what basis the authors assign various levels to water systems and catchments.
3. On what basis the authors provided cooling ecological nodes in the study area.
4. The authors focused on the path of water corridors forgetting its width.
5. Why the authors did not take into account temporal temperature variability.
6. How the authors superpose the water systems and temperature watersheds to get cooling ecological nodes.
Reviewer 2 Report
Initially, I would like to congratulate the authors for the excellent work submitted to the journal. The importance of discussing the discussed topic is of unique relevance to environmental science as a whole. However, the work is lacking in some aspects, which end up weakening it too much.
In the presentation of the study area, it would be interesting to include some photos of the city that are of importance for the discussion of the results.
What is the prevailing wind direction in this city? And the average speed? It would be interesting to include climatic data from the study area in its presentation, so that readers have an idea of ​​which direction the wind can cool the hot spots of the city.
In all tables: organize the data to the right, respecting the position of the decimal places.
Very interesting research, congratulations to the authors.
Round 2
Reviewer 2 Report
Authors now have provided the final version with the suggestions included. This is an important paper for urban thermal environmental studies. Congratulations. I agree wiht the publication.